# Prescription Patterns of Drugs Given to Hospitalized COVID-19 Patients: A Cross-Sectional Study in Colombia

**DOI:** 10.3390/antibiotics11030333

**Published:** 2022-03-03

**Authors:** Luis Fernando Valladales-Restrepo, Jaime Andrés Giraldo-Correa, Brayan Stiven Aristizábal-Carmona, Camilo Alexander Constain-Mosquera, Alejandra Sabogal-Ortiz, Jorge Enrique Machado-Alba

**Affiliations:** 1Grupo de Investigación en Farmacoepidemiología y Farmacovigilancia, Universidad Tecnológica de Pereira-Audifarma S.A., Pereira 660004, Colombia; lfvalladales@utp.edu.co; 2Grupo de Investigación Biomedicina, Facultad de Medicina, Fundación Universitaria Autónoma de las Américas, Pereira 660004, Colombia; 3Semillero de Investigación en Farmacología Geriátrica, Grupo de Investigación Biomedicina, Facultad de Medicina, Fundación Universitaria Autónoma de las Américas, Pereira 660004, Colombia; jaime.giraldoc@uam.edu.co (J.A.G.-C.); brayan.aristizabal@uam.edu.co (B.S.A.-C.); camilo.constain@uam.edu.co (C.A.C.-M.); 4Grupo Ospedale, Operador Clínico Hospitalario por Outsourcing S.A.S—G—Ocho S.A.S, Área de Salud, Cali 760009, Colombia; asesormodelogr@ospedale.com.co

**Keywords:** COVID-19, SARS-CoV-2, corticosteroids, azithromycin, ivermectin, colchicine, pharmacoepidemiology

## Abstract

The impact of COVID-19 prompted a race to find a treatment that would reduce its mortality. Most studies have not shown favorable results for many of these drugs, but they are still used. The aim as to determine the differences and similarities in the hospital pharmacological management of patients with COVID-19 according to sex, age group, and geographical region of Colombia, 2020–2021. Descriptive cross-sectional study was conducted on the prescription patterns of the medications given to patients diagnosed with COVID-19 treated in eight clinics in Colombia between 6 March 2020 and 31 May 2021. We performed a descriptive analysis of the sociodemographic, clinical, and pharmacological variables of the patients. A total of 8596 patients from 170 cities were identified, with a median age of 53.0 years and 53.3% of them men. A total of 24.3% required care in the intensive care unit, and 18.7% required invasive mechanical ventilation. The most commonly used drugs for the treatment of COVID-19 were systemic corticosteroids (63.6%), followed by colchicine (12.8%), azithromycin (8.9%), and ivermectin (6.4%). Corticosteroids, anticoagulants, colchicine, azithromycin, ivermectin, and hydroxychloroquine were prescribed more frequently in men, and their overall use increased with age. There were differences in prescriptions between geographic regions. The majority of patients were managed with medications included in the management guidelines. There were differences between sexes, age groups, and geographical regions.

## 1. Introduction

The type 2 coronavirus that causes severe acute respiratory syndrome (SARS-CoV-2) is the cause of the 2019 coronavirus disease (COVID-19) [1], which has had a great global impact as a severe public health problem, such that on 12 March 2020, the World Health Organization declared it a pandemic [2]. More than 238 million cases of COVID-19 have been reported, with an approximate mortality of 2.0% worldwide [3]. In Colombia, according to the National Institute of Health, almost 5 million people have contracted and more than 126 thousand people have died from COVID-19 [4]. This pandemic has generated an unprecedented burden on health systems around the world in the form of a greater number of hospital admissions and high demands for intensive care unit (ICU) beds, advanced respiratory support, renal replacement therapy, and other interventions for life support and medical care [5]. The impact of the COVID-19 pandemic on the health system of each country has been different, depending on the balance between the supply and demand of services, which has been associated with the ability to expand the number of hospital beds, in particular in the ICU, and public health policies to contain the pandemic [5,6].

In the midst of the need to find an effective treatment for COVID-19, antiviral drugs, antibiotics, antimalarials, immunosuppressants, immunomodulators, and corticosteroids, among others, have been chosen as therapeutic measures that help reduce the mortality of mechanical ventilation, severe acute respiratory syndrome (SARS), and other complications caused by the virus [7,8]. Many of the proposed drugs have a possible mechanism of action on SARS-CoV-2 or on the pathophysiology of the disease [9,10]. However, most drugs have not demonstrated efficacy through clinical studies, and others still do not have enough evidence to be recommended [7,8]. Even so, they are used in the general population, bringing the risk of drug interactions and adverse drug reactions [11,12]. 

The objective of this study was to determine the differences and similarities in the hospital pharmacological management of patients with COVID-19 according to sex, age group, and geographical region of Colombia.

## 2. Results

A total of 8596 patients from 170 different cities were identified. About half (53.3%, n = 4583) were men, and the median age was 53.0 years (interquartile range: 37.0–68.0 years; range: 0.0–101.0 years). A total of 2.1% (n = 179) were younger than 18 years, 26.0% (n = 2236) were between 18 and 39 years, 40.2% (n = 3457) were between 40 and 64 years, and 29.6% (n = 2541) were 65 or older. The age of 2.1% (n = 183) of the patients was unknown. In order of frequency, patients were from the Pacific region (n = 2718; 31.6%), followed by the Central region (n = 2403; 28.0%), Caribbean region (n = 1756; 20.4%), and Bogotá-Cundinamarca region (n = 1719; 20.0%). A total of 42.5% (n = 3653) had some chronic comorbidity, the most frequent being arterial hypertension (n = 2785; 32.4%), followed by diabetes mellitus (n = 1167; 13.6%), obstructive pulmonary disease (n = 326; 3.8%), chronic kidney disease (n = 291; 3.4%), and dyslipidemia (n = 263; 3.1%).

Symptoms were reported in 4804 patients, most often cough (n = 4018/4804; 83.6%), fever (n = 2879; 59.9%), odynophagia (n = 2186; 45.5%), dyspnea (n = 2179; 45.4%), and fatigue (n = 2141; 44.6%). The majority of cases required the highest level of medical care in the general hospitalization wards (n = 3457; 40.2%), followed by emergency services (n = 3050; 35.5%) and ICUs (n = 2089; 24.3%). A total of 56.3% (n = 4842) required supplemental oxygen, and 18.7% (n = 1605) required invasive mechanical ventilation.

The medications used for the management of SARS-CoV-2 infection were most often corticosteroids (n = 5465; 63.6%), especially dexamethasone (n = 5095; 59.3%) and methylprednisolone (n = 893; 10.4%). Anticoagulants were used in 61.7% (n = 5300) of cases, especially enoxaparin (n = 5049; 58.7%) and unfractionated heparin (n = 995; 11.6%). Other prescribed medications were colchicine (n = 1098; 12.8%), azithromycin (n = 761; 8.9%), ivermectin (n = 549; 6.4%), hydroxychloroquine (n = 66; 0.8%), chloroquine (n = 48; 0.6%), lopinavir/ritonavir (n = 15; 0.2%), and tocilizumab (n = 9; 0.1%). The most frequently found comedications were analgesics with anti-inflammatory drugs (n = 6214; 72.3%), antiulcer drugs (n = 5079; 59.1%), systemic antibiotics (n = 4919; 57.2%), antihypertensives with diuretics (n = 3076; 35.8%), and hypnotics with sedatives (n = 1951; 22.7%). Vasopressors and inotropes were used in 18.5% of patients (n = 1587), and muscle relaxants were used in 17.7% (n = 1525). A total of 15.0% (n = 1292) of the patients died.

### Comparisons of Drug Use

Systemic corticosteroids, anticoagulants, colchicine, azithromycin, ivermectin, and hydroxychloroquine were prescribed significantly more often to men than women (Table 1). In general, the proportion of medications used for the management of COVID-19 increased with age, especially after 40 years of age. The antimalarials lopinavir/ritonavir and tocilizumab were not prescribed in children under 18 years of age (Table 2). Corticosteroids were prescribed mainly in the Pacific and Central regions. Colchicine was prescribed in the Pacific, Central, and Caribbean regions. Azithromycin, ivermectin, and chloroquine were prescribed with greater frequency in the Pacific region (Table 3). Table 1, Table 2 and Table 3 show the comparisons of other sociodemographic, clinical, and pharmacological variables according to sex, age group, and geographic region.

## 3. Discussion

This analysis allowed us to characterize the hospital pharmacological treatment of a group of patients with a confirmed diagnosis of COVID-19 treated in eight cities of Colombia. In general, most of the findings of this report regarding sociodemographic data, comorbidities and pharmacological treatment were consistent with what has been described in other studies carried out in the world [13,14,15,16,17,18,19,20,21,22]. The median age was similar to that found in other studies (55.0–60.0 years) [13,14,23] and lower than that reported by others (50.4–68.1 years) [19,20,21,22,24], with a predominance of men, as identified in the majority of studies [13,14,20,22,23,24,25,26], which may be due to biological factors such as genes, sex hormones and microbiome that may influence the host immune responses to infections [27]; though some reports have had a higher proportion of women (51.2–64.6%) [15,16,28]. The most frequently found comorbidities were arterial hypertension, diabetes mellitus, and chronic obstructive pulmonary disease, in line with other studies [13,14,20,21,22,23,24,29]. These are pathologies that have been associated with a greater probability of presenting complications, severe forms of the disease and death [30,31]. For this reason, almost one-fifth of the patients required invasive mechanical ventilation, consistent with other reports [32,33], and approximately a quarter needed to be managed in the ICU, a proportion that was similar to that found in Greece (20.0%) [32] and lower than that described in other Colombian studies (32.4–47.8%) [14,33,34].

Most patients received some systemic corticosteroids, similar to earlier reports (56.5–69.7%) [15,16,17,23,34] but contrasting with others that prescribed them less often [19,24,25]. The most often prescribed one was dexamethasone, consistent with what was found in the United States and Colombia [16,24,33,34] but different from what was found in Spain and Peru, where the use of methylprednisolone predominated [17,25,28], and in Pakistan, where hydrocortisone predominated [26]. Its use is based on the findings of the RECOVERY study, which showed that dexamethasone reduced the risk of death by 36% in patients with invasive mechanical ventilation and by 18% among patients who required supplemental oxygen [35]. Dexamethasone is currently recommended by different international [36,37,38] and national clinical practice guidelines for patients [39], especially those with hypoxemia or with oxygen or mechanical ventilation requirements [36,37,38,39]. Although dexamethasone has been shown to be effective in these patients, some serious side effects are associated with its use, such as hyperglycemia, fluid retention, weight gain, bacterial superinfection, confusion, and behavioral changes, so its use must be weighed against the benefits and risks [36,37,38,39]. Another therapeutic group with strong evidence of their worth in patients hospitalized for COVID-19 has been anticoagulants [37,39], used in more than half of the patients in this report, a finding consistent with several studies (50.4–75.0%) [14,16].

Tocilizumab is another drug that has growing evidence in favor of its use [36,37]. In this report, it was prescribed in a small portion of the patients, contrasting with the 3.1–10.8% in other studies [13,15,18,24,28]. Some guidelines recommend it in combination with systemic corticosteroids for hospitalized adults who have rapid respiratory decompensation due to COVID-19 [36,37]. The last update of the Colombian consensus did not give a recommendation for or against the drug due to its cost, its limited availability, and its lack of current approval by the National Institute of Food and Drug Surveillance (INVIMA, for its name in Spanish) for said indication [39], which explains its low use. On the other hand, remdesivir is approved for use in certain patients [36,37], but it is not available in Colombia [39]. Likewise, tofacitinib may be recommended for some adults hospitalized for severe COVID-19 but without mechanical ventilation [36], while other guidelines do not recommend this medication [37,39], which helps explain its absence in this cohort of patients.

Colchicine, an anti-inflammatory agent that is approved for gout, recurrent pericarditis, Behçet’s disease, and familial Mediterranean fever [40], is not recommended for use in patients hospitalized for COVID-19 [37,39], and in other clinical practice guidelines, it is not even considered a treatment [36,38]. However, it was found to be prescribed in more than a tenth of the present patients with COVID-19, which is consistent with what was found in Greece (8.2%) [32] and Peru (14.3%) [18] but is notably lower than its prescription rate in two other studies conducted in Colombia (48.2–54.1%) [14,33] because the drug was recommended in the management protocols of the institutions involved in the studies [14,33]. As previously mentioned, this drug is not recommended for the management of patients with COVID-19, but preliminary data from some meta-analyses have shown that colchicine can reduce mortality, although most of the studies included in the analyzes were observational. Therefore, clinical trials are required to corroborate these findings [41,42,43].

Mainly at the beginning of the pandemic, other drugs were also used without solid scientific knowledge but that sought to impact the morbidity and mortality of patients. The evidence was increasingly robust, showing that medications such as azithromycin, chloroquine, hydroxychloroquine, lopinavir/ritonavir, and ivermectin did not offer benefit in the treatment of SARS-CoV-2 infection and could even be related to a higher risk of pharmacological interactions and adverse events such as QT interval prolongation, arrhythmias, elevated liver enzymes, blood dyscrasias, seizures, skin rash [36,37,38,39]. Thus, there are reports of studies from different countries that show a wide use of azithromycin (60.6–88.6%) [13,25,26,28], which contrasts with the low use found in this report. Likewise, ivermectin was prescribed in 6.4% of our patients, contrasting with that found in Pakistan (2.2%) [26] and Peru (37.0%) [18]. Some of these prescriptions could be founded in sound prophylaxis to avoid hyperinfection by *Strongyloides stercoralis* when given before the use of corticosteroids [39]. Finally, antimalarials and lopinavir/ritonavir were used in fewer than 2% of patients, which contrasts significantly with other studies [13,17,25,28,32,34,44].

There were evident differences in the prescription of these drugs between men and women, by age group, and by geographical region of the country. This dynamic is not unheard of [15,24]. For example, in two studies conducted in the US, Stroever et al. found that the prescription of corticosteroids predominated in women, and the use of medications such as hydroxychloroquine, tocilizumab, lopinavir/ritonavir, and corticosteroids increased with age up to 65 years. The patterns of use of these drugs varied according to geographical area [15]. Best et al. found such differences in terms of geographic region and age group, and differences were also found depending on race or ethnic group [24]. Several factors can influence the differences in drug use patterns, such as medical decisions, which are affected by demographic, cultural, and economic aspects, as well as the academic training of the doctor. In addition, the differences between countries may reflect the availability or absence of the drug, its costs, the approved indications, the characteristics of the patients, the health system itself, the local management guidelines, and the influence of the media [45,46].

Some limitations should be considered when interpreting our results. First, we did not have access to medical records to verify all pathologies and complications during hospitalization, so we could not verify the accuracy of the diagnoses assigned by the doctor, nor the severity of COVID-19, among other clinical and paraclinical variables. Similarly, any medications prescribed outside the health system or not delivered by the dispensing company are unknown. One strength is that the study included a significant number of patients distributed in most geographic regions of the national territory.

## 4. Materials and Methods

A cross-sectional study was conducted on the prescription patterns of drugs given to hospitalized patients with a diagnosis of COVID-19. These were identified from the report of confirmed positive cases by polymerase chain reaction or antigen reactivity performed by the Ospedale Group network, which consists of eight clinics located in the cities of Armenia, Barranquilla, Bogotá, Cartagena, Cali, Manizales, Pereira, and Popayán.

From this population, patients with a first confirmed diagnosis of COVID-19 were selected, of any age, sex, and city of residence, between 6 March 2020 and 31 May 2021, who were treated in-hospital in emergency services, in the general ward, or in the ICU (Figure 1). With this identified patient list, information on the use of medications was obtained from the dispensing company (Audifarma S.A.). A database was set up to gather the following patient variables:1.Sociodemographic: sex, age (<18 years, 18–39 years, 40–64 years, 65 or more years), and city. The place of care was categorized by department and region of Colombia, taking into account the classification of the National Administrative Department of Statistics (DANE) of Colombia, as follows:Caribbean region: Atlántico (Barranquilla), Bolívar (Cartagena);Central region: Caldas (Manizales), Quindío (Armenia), Risaralda (Pereira);Bogotá-Cundinamarca region: Bogotá;Pacific region: Cauca (Popayán), Valle del Cauca (Cali).2.Clinical: comorbidities (arterial hypertension, chronic obstructive pulmonary disease, obesity, dyslipidemia, diabetes mellitus, depression, anxiety, chronic kidney disease, asthma, heart failure, ischemic heart disease, among others) and clinical manifestations (cough, dyspnea, fever, fatigue, odynophagia, precordial pain, asthenia/adynamia, among others).3.Place of care: emergency room, general ward, or ICU.4.Supplemental oxygen: oxygen requirement, mechanical ventilation, and need for tracheostomy.5.Medications that have been used in patients with COVID-19 include systemic corticosteroids (dexamethasone, hydrocortisone, methylprednisolone, prednisolone, betamethasone, prednisone), anticoagulants (unfractionated heparin, low molecular weight heparins), antimalarials (chloroquine, hydroxychloroquine), ivermectin, lopinavir/ritonavir, colchicine, tocilizumab, nitazoxanide, tofacitinib and remdesivir (not available in the country).6.Comedications: they were grouped into the following categories: (a) antidiabetics, (b) antihypertensives and diuretics, (c) lipid-lowering drugs; (d) antiulcer drugs, (e) antidepressants, (f) anxiolytics and hypnotics, (g) thyroid hormone, (h) antipsychotics, (i) antiepileptics, (j) antiarrhythmics, (k) antihistamines, (l) antiplatelets, (m) analgesics and anti-inflammatories, (n) systemic antibiotics, (o) bronchodilators and inhaled corticosteroids, (p) vasopressors and inotropics, (q) muscle relaxants, (r) hypnotics and sedatives, and (s) others.

The protocol was endorsed by the Bioethics Committee of the Technological University of Pereira in the category of “research without risk” (approval code: 30-070421). The principles of confidentiality of information established by the Declaration of Helsinki were respected.

### Statistical Analysis

The data were analyzed with the statistical package SPSS Statistics, version 26.0, for Windows (IBM, USA). A descriptive analysis was performed by calculating frequencies and proportions for the qualitative variables and measures of central tendency and dispersion for the quantitative variables, depending on their parametric behavior as shown by the Kolmogorov–Smirnov test. Quantitative variables were compared by the Mann–Whitney U test and the χ*^2^* or Fisher’s exact test for categorical variables. The level of statistical significance was *p* < 0.05.

## 5. Conclusions 

From the above results, we can conclude that the majority of patients with a confirmed diagnosis of COVID-19 were managed with medications included in the management guidelines. The prescriptions differed by sex, age, and the geographic regions where they were treated. These findings can be useful for clinicians who treat these patients and for decision-makers to strengthen continuing education programs for physicians to optimize the quality of their prescriptions and thereby improve the quality of care and reduce the risks to patients.

## Figures and Tables

**Figure 1 antibiotics-11-00333-f001:**
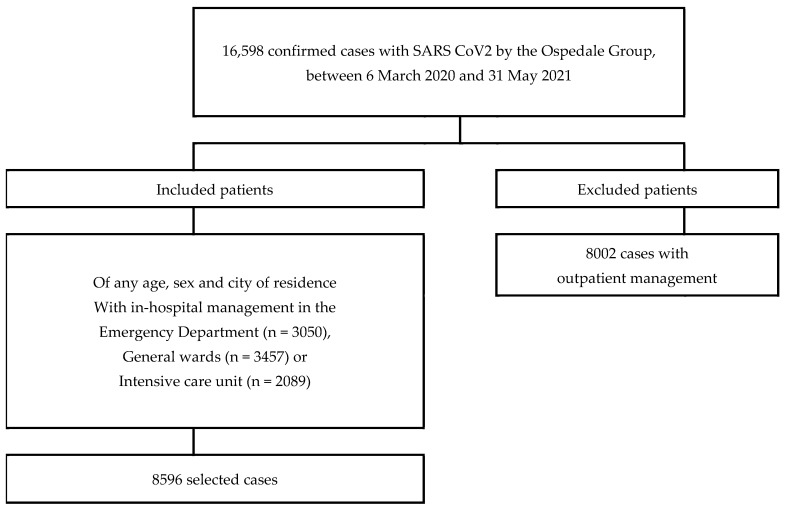
Flowchart with the determination of patients who were included with pharmacological treatment for Covid-19 in different regions of Colombia.

**Table 1 antibiotics-11-00333-t001:** Comparison of some sociodemographic, clinical and pharmacological variables by sex, in a group of patients with a confirmed diagnosis of SARS-CoV-2 infection, Colombia.

Variables	Men		Women		*p*
	n = 4583	%	n = 4013	%	
Age, median (IQR)	55.6 (41.0–69.0)		50.2 (33.0–67.0)		<0.001 *
Geographic region					
Pacific region	1588	34.6	1130	28.2	<0.001
Central region	1218	26.6	1185	29.5	0.002
Caribbean region	949	20.7	807	20.1	0.493
Bogota-Cundinamarca region	828	18.1	891	22.2	<0.001
Comorbidities					
Arterial hypertension	1483	32.4	1302	32.4	0.932
Diabetes mellitus	646	14.1	521	13.0	0.133
Chronic obstructive pulmonary disease	186	4.1	140	3.5	0.168
Chronic kidney disease	177	3.9	114	2.8	0.009
Dyslipidemia	139	3.0	124	3.1	0.878
Place of care					
Emergency department or observation	1436	31.3	1614	40.2	<0.001
Hospitalization in general wards	1819	39.7	1638	40.8	0.288
Intensive care unit	1328	29.0	761	19.0	<0.001
Treatment					
Supplemental oxygen	2879	62.8	1963	48.9	<0.001
Invasive mechanical ventilation	998	21.8	607	15.1	<0.001
Tracheostomy	54	1.2	17	0.4	<0.001
Corticosteroids	3107	67.8	2358	58.8	<0.001
Dexamethasone	2916	63.6	2179	54.3	<0.001
Methylprednisolone	521	11.4	372	9.3	0.001
Hydrocortisone	244	5.3	146	3.6	<0.001
Betamethasone	84	1.8	62	1.5	0.303
Prednisolone	40	0.9	42	1.0	0.408
Prednisone	2	0.0	44	1.1	<0.001
Parenteral anticoagulants	3151	68.8	2149	53.6	<0.001
Low molecular weight heparins	3049	66.5	2047	51.0	<0.001
Unfractionated heparin	600	13.1	395	9.8	<0.001
Colchicine	670	14.6	428	10.7	<0.001
Azithromycin	513	11.2	248	6.2	<0.001
Ivermectin	351	7.7	198	4.9	<0.001
Hydroxychloroquine	46	1.0	20	0.5	0.007
Chloroquine	28	0.6	20	0.5	0.485
Lopinavir-Ritonavir	9	0.2	6	0.1	0.603
Tocilizumab	7	0.2	2	0.0	0.188 **
Comedications					
Analgesics-anti-inflammatories	3301	72.0	2913	72.6	0.561
Antiulcer	2968	64.8	2111	52.6	<0.001
Systemic antibiotics	2903	63.3	2016	50.2	<0.001
Antihypertensives and diuretics	1771	38.6	1305	32.5	<0.001
Hypno-sedatives	1181	25.8	770	19.2	<0.001
Mortality	812	17.7	480	12.0	<0.001

IQR: Interquartile range; * Mann–Whitney U test. ** Fisher’s test.

**Table 2 antibiotics-11-00333-t002:** Comparison of some sociodemographic, clinical and pharmacological variables by age, in a group of patients with a confirmed diagnosis of SARS-CoV-2 infection, Colombia.

Variables	<18 Years		18–39 Years		40–64 Years		≥65 Years	
	n = 179	%	n = 2236	%	n = 3457	%	n = 2541	%
Men	77	43.0	949	42.4	2031	58.8	1440	56.7
Geographic region								
Pacific region	35	19.6	458	20.5	1159	33.5	1061	41.8
Central region	36	20.1	703	31.4	826	23.9	670	26.4
Caribbean region	53	29.6	503	22.5	750	21.7	447	17.6
Bogota-Cundinamarca region	55	30.7	572	25.6	722	20.9	363	14.3
Comorbidities								
Arterial hypertension	9	5.0	245	11.0	1060	30.7	1462	57.5
Diabetes mellitus	3	1.7	67	3.0	474	13.7	617	24.3
Chronic obstructive pulmonary disease	0	0.0	3	0.1	47	1.4	268	10.5
Chronic kidney disease	0	0.0	13	0.6	100	2.9	178	7.0
Dyslipidemia	0	0.0	20	0.9	150	4.3	93	3.7
Place of care								
Emergency department or observation	97	54.2	1287	57.6	1185	34.3	456	17.9
Hospitalization in general wards	57	31.8	763	34.1	1413	40.9	1085	42.7
Intensive care unit	25	14.0	186	8.3	859	24.8	1000	39.4
Treatment								
Supplemental oxygen	47	26.3	585	26.2	2072	59.9	2027	79.8
Invasive mechanical ventilation	10	5.6	138	6.2	624	18.1	792	31.2
Tracheostomy	0	0.0	7	0.3	20	0.6	42	1.7
Corticosteroids	42	23.5	974	43.6	2327	67.3	1995	78.5
Dexamethasone	29	16.2	870	38.9	2198	63.6	1872	73.7
Methylprednisolone	8	4.5	120	5.4	336	9.7	405	15.9
Hydrocortisone	6	3.4	53	2.4	157	4.5	165	6.5
Betamethasone	1	0.6	20	0.9	51	1.5	70	2.8
Prednisolone	0	0.0	14	0.6	28	0.8	36	1.4
Prednisone	1	0.6	40	1.8	3	0.1	1	0.0
Parenteral anticoagulants	12	6.7	626	28.0	2330	67.4	2218	87.3
Low molecular weight heparins	12	6.7	608	27.2	2267	65.6	2119	83.4
Unfractionated heparin	2	1.1	64	2.9	322	9.3	556	21.9
Colchicine	1	0.6	101	4.5	503	14.6	440	17.3
Azithromycin	3	1.7	70	3.1	372	10.8	314	12.4
Ivermectin	3	1.7	63	2.8	247	7.1	236	9.3
Hydroxychloroquine	0	0.0	12	0.5	37	1.1	17	0.7
Chloroquine	0	0.0	8	0.4	30	0.9	10	0.4
Lopinavir-Ritonavir	0	0.0	4	0.2	7	0.2	4	0.2
Tocilizumab	0	0.0	2	0.1	5	0.1	2	0.1
Comedications								
Analgesics-anti-inflammatories	91	50.8	1649	73.7	2600	75.2	1733	68.2
Antiulcer	31	17.3	724	32.4	2176	62.9	2038	80.2
Systemic antibiotics	68	38.0	699	31.3	2102	60.8	1956	77.0
Antihypertensives and diuretics	10	5.6	217	9.7	1129	32.7	1653	65.1
Hypno-sedatives	14	7.8	200	8.9	750	21.7	937	36.9
Mortality	3	1.7	52	2.3	418	12.1	810	31.9

**Table 3 antibiotics-11-00333-t003:** Comparison of some sociodemographic, clinical and pharmacological variables by geographic regions, in a group of patients with a confirmed diagnosis of SARS-CoV-2 infection, Colombia.

Variables	Pacific Region		Central Region		Caribbean Region		Bogota-Cundinamarca Region	
	n = 2718	%	n = 2403	%	n = 1756	%	n = 1719	%
Age, median (IQR)	59.0 (44.0–72.0)		51.0 (34.0–68.0)		51.0 (36.0–65.0)		47.0 (32.0–62.0)	
Men	1588	58.4	1218	50.7	949	54.0	828	48.2
Comorbidities								
Arterial hypertension	852	31.3	864	36.0	660	37.6	409	23.8
Diabetes mellitus	405	11.9	315	13.1	290	16.5	157	9.1
Chronic obstructive pulmonary disease	114	4.2	138	5.7	30	1.7	44	2.6
Chronic kidney disease	121	4.5	59	2.5	73	4.2	38	2.2
Dyslipidemia	80	2.9	113	4.7	41	2.3	29	1.7
Place of care								
Emergency department or observation	774	28.5	713	29.7	408	23.2	1155	67.2
Hospitalization in general wards	949	34.9	1330	55.3	835	47.6	343	20.0
Intensive care unit	995	36.6	360	15.0	513	29.2	221	12.9
Treatment								
Supplemental oxygen	1859	68.4	1304	54.3	900	51.3	779	45.3
Invasive mechanical ventilation	657	24.2	465	19.4	289	16.5	194	11.3
Tracheostomy	50	1.8	18	0.7	3	0.2	0	0.0
Corticosteroids	1865	68.6	1600	66.6	1048	59.7	952	55.4
Dexamethasone	1772	65.2	1519	63.2	883	50.3	921	53.6
Methylprednisolone	294	10.8	329	13.7	223	12.7	47	2.7
Hydrocortisone	146	5.4	103	4.3	115	6.5	26	1.5
Betamethasone	2	0.1	13	0.5	23	1.3	8	0.5
Prednisolone	34	1.3	88	3.7	6	0.3	18	1.0
Prednisone	35	1.3	38	1.3	3	0.2	6	0.3
Parenteral anticoagulants	1991	73.3	1337	55.6	1125	64.1	847	49.3
Low molecular weight heparins	1939	71.3	1219	50.7	1116	63.6	822	47.8
Unfractionated heparin	512	18.8	357	14.9	37	2.1	89	5.2
Colchicine	457	16.8	374	15.6	240	13.7	27	1.6
Azithromycin	690	25.4	9	0.4	62	3.5	0	0.0
Ivermectin	483	17.8	49	2.0	15	0.9	2	0.1
Hydroxychloroquine	17	0.6	4	0.2	40	2.3	5	0.3
Chloroquine	38	1.4	9	0.4	1	0.1	0	0.0
Lopinavir-Ritonavir	6	0.2	6	0.2	3	0.2	0	0.0
Tocilizumab	4	0.1	0	0.0	5	0.3	0	0.0
Comedications								
Analgesics-anti-inflammatories	2013	74.1	1841	76.6	1208	68.8	1152	67.0
Antiulcer	1999	73.5	1390	57.8	1036	59.0	654	38.0
Systemic antibiotics	1727	63.5	1241	51.6	1239	70.6	712	41.4
Antihypertensives and diuretics	1242	45.7	865	36.0	552	31.4	417	24.3
Hypno-sedatives	799	29.4	575	23.9	363	20.7	214	12.4
Mortality	567	20.9	196	8.2	314	17.9	215	12.5

IQR: Interquartile range.

## Data Availability

The database is available at protocols.io. Data access: dx.doi.org/10.17504/protocols.io.b2dnqa5e.

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
