# Peer review of "Prescription Patterns of Drugs Given to Hospitalized COVID-19 Patients: A Cross-Sectional Study in Colombia"

_antibiotics, 2022, doi:10.3390/antibiotics11030333_

Round 1
Reviewer 1 Report
This is an interesting manuscript. But, several concerns need to be addressed to fit for publication as follows:
1. The title seems like a review title. I think the following title could be more representative " The prescription patterns of drugs given to hospitalized COVID-19 patients: A cross-sectional study in Colombia".
2. Abstract: line 21 "Descriptive cross-sectional study" seems an incomplete sentence.
3. Table 2: what is the meaning of "años".
4. The discussion needs much improvement as the authors should clarify the possible interpretations of their findings not just comparing with the earlier studies. For example Line 120: "with a predominance of men". Mention the possible reasons for the earlier predominance. Also, in line 153" colchicine is not recommended for use in patients hospitalized for COVID-19" while in line 158", Different meta-analyses show that colchicine reduces mortality in patients with COVID-19". The authors should clarify such discrepancy
5. Add a subheading entitled "Statistical analysis" before line 229.
Author Response
Colombia, Feb 10 of 2022
Antibiotics
Reference: antibiotics-1586856
Dear editors, we respond to each of the comments. Thank you very much for your help in improving the manuscript.
Reviewer 1:
- The title seems like a review title. I think the following title could be more representative " The prescription patterns of drugs given to hospitalized COVID-19 patients: A cross-sectional study in Colombia".
Answer: The change is made.
- Abstract: line 21 "Descriptive cross-sectional study" seems an incomplete sentence.
Answer: The sentence is completed: “Descriptive cross-sectional study was conducted on the prescription patterns of the medications given to patients diagnosed with COVID-19 treated in eight clinics in Colombia between March 6, 2020 and May 31, 2021”
- Table 2: what is the meaning of "años".
Answer: “años” is changed for years
- The discussion needs much improvement as the authors should clarify the possible interpretations of their findings not just comparing with the earlier studies. For example, Line 120: "with a predominance of men". Mention the possible reasons for the earlier predominance. Also, in line 153" colchicine is not recommended for use in patients hospitalized for COVID-19" while in line 158", Different meta-analyses show that colchicine reduces mortality in patients with COVID-19". The authors should clarify such discrepancy
Answer: Settings are made
“with a predominance of men, as identified in the majority of studies (13-15, 17, 19-22), which may be due to biological factors such as genes, sex hormones and microbiome that may influence the host immune responses to infections (23); though some reports have had a higher proportion of women (51.2-64.6%) (24-26)”
As previously mentioned, this drug is not recommended for the management of patients with COVID-19, but preliminary data from some meta-analyses have shown that colchicine can reduce mortality, although most of the studies included in the analyzes were observational. Therefore, clinical trials are required to corroborate these findings (40-42).
- Add a subheading entitled "Statistical analysis" before line 229.
Answer: The subheading is added
The authors

Reviewer 2 Report
The article “COVID-19: Hospital pharmacological treatment according to geographic region in Colombia” describes the comparison of hospital pharmacological management of patients with COVID-19 according to the sex, age group and geographical region of Colombia during the period of 2020-2021. Descriptive analysis of the sociodemographic, clinical, and pharmacological variables in huge number of the patients indicates that there were differences in prescriptions between geographic regions. This study concludes that there were differences between sexes, age groups, and geographical regions. Overall the article is presented well. I have a few suggestions to improve the quality of this manuscript.
Comments
- The authors can make a general statement of what is observed here is comparable with other parts of the worldwide.
- A flow chart showing study profile could be included to have a clear picture of the study design.
- The discussion could be improved by signifying the limitations of each therapy, which could be more interesting for the readers.
Author Response
Colombia, Feb 10 of 2022
Antibiotics
Reference: antibiotics-1586856
Dear editors, we respond to each of the comments. Thank you very much for your help in improving the manuscript.
Reviewer 2:
The article “COVID-19: Hospital pharmacological treatment according to geographic region in Colombia” describes the comparison of hospital pharmacological management of patients with COVID-19 according to the sex, age group and geographical region of Colombia during the period of 2020-2021. Descriptive analysis of the sociodemographic, clinical, and pharmacological variables in huge number of the patients indicates that there were differences in prescriptions between geographic regions. This study concludes that there were differences between sexes, age groups, and geographical regions. Overall the article is presented well. I have a few suggestions to improve the quality of this manuscript.
Comments
- The authors can make a general statement of what is observed here is comparable with other parts of the worldwide.
Answer: we add in first paragraph of discussion “In general, most of the findings of this report regarding sociodemographic data, comorbidities and pharmacological treatment were consistent with what has been described in other studies carried out in the world (13-22)”
- A flow chart showing study profile could be included to have a clear picture of the study design.
Answer: The flowchart is made (see figure 1).
- The discussion could be improved by signifying the limitations of each therapy, which could be more interesting for the readers.
Answer: Limitations such as interactions and adverse reactions are added:
“Although dexamethasone has been shown to be effective in these patients, some serious side effects are associated with its use, such as hyperglycemia, fluid retention, weight gain, bacterial superinfection, confusion, and behavioral changes, so its use must be weighed against the benefits and risks (36-39)”
“As previously mentioned, this drug is not recommended for the management of patients with COVID-19, but preliminary data from some meta-analyses have shown that colchicine can reduce mortality, although most of the studies included in the analyzes were observational. Therefore, clinical trials are required to corroborate these findings (41-43)”
“such as QT interval prolongation, arrhythmias, elevated liver enzymes, blood dyscrasias, seizures, skin rash, etc (36-39)”
The authors

Round 2
Reviewer 1 Report
The authors addressed all comments